# The Celestial Masters and the Origins of Daoist Monasticism

## Qi Sun

Advanced Institute of Confucian Studies, Shandong University, Jinan 250100, China; sun_qi@sdu.edu.cn

**Abstract:** The Daoist monasteries, which were first popularized in southern China in the late fifth century, reflected major changes in the structure of medieval Daoism. From the perspective of comparative religious history, the rise of Daoist monasteries bears some similarity to the monasticisms that came into being in the Christian and Buddhist traditions; all three originated in hermitic and ascetic practices. However, Daoist monasticism did not naturally stem from the hermetic Daoism tradition; instead, it underwent a two-stage process of "grafting" in terms of its spiritual beliefs and values. The first stage saw the emergence of Daoist scriptures in the Jin and Song periods; in particular, the Lingbao scriptures, which transformed and distilled the tradition of hermetic Daoism practiced in the mountains and invested hermitic practice with a more complete and sacrosanct doctrinal foundation. The second saw the Southern Dynasties' Celestial Masters order embrace and experiment with the beliefs and values within the Lingbao scriptures; this process introduced the inherent communitarian nature of the Celestial Masters into the development of Daoist monasticism and resulted in the large-scale transformation of religious practice among the Celestial Masters of the period. This change of direction among the Celestial Masters order in the Jin and Song periods toward mountain-based practice led to the establishment of Daoist monasticism, but also to a loss of purity therein.

**Keywords:** Daoist monasticism; Daoist monasteries; Daoguan; Celestial Masters



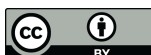

## 1. Introduction

The appearance of Daoist monasteries (*Daoguan*, 道觀 or 道館) in the fifth century was a key turning point in the history of Daoism. Not only did the appearance and popularization of Daoist monasteries reshape the cultural landscape in the mountain regions of ancient China, but these events also brought major changes to Daoism's social structure. Daoism transformed from a religious community rooted in rural society to a monastic religion of renunciation and asceticism (Wei 2019; Sun 2020).

Scholars generally mark the establishment of the Chongxu Guan Temple 崇虛館 in Nanjing (then known as Jiankang 建康) in the third year of the Tai Shi 泰始 reign (467) in the Southern Dynasties as the beginning of the history of Daoist monasteries. In that same period, at least four Daoist temples appeared all at once across southeast China (Sun 2014, pp. 158–67). It is difficult to understand why Daoist monasteries would appear so abruptly. Scholars have worked assiduously to trace the monasteries' origins. It has been suggested that virtually all the pre-existing spaces devoted to Daoist practice are connected with the rise of Daoist monasteries. Such spaces include the temple buildings (*jingshe* 精舍) of the mountain-dwelling hermits of the Wei (220–266) and Jin (265–420) dynasties, the "chambers of quietude" (*jingshi* 靖室) of Daoist families, the "halls of parishes" (*zhitang* 治堂) within the homes of the Celestial Masters order (*tianshi dao* 天師道) practitioners, the Mao Mountain 茅山 villa belonging to the Shangqing 上清 school, the "abstinence halls" (*zhaitang* 齋堂) depicted in early Lingbao 靈寶 scriptures, and the guesthouses arranged for hermits (Kohn 2000; Bumbacher 2000, pp. 490–93; Kohn 2003; Akiko 2009; Wang 2017, pp. 3–171; Wei 2017).

However, the reasons underlying the rapid spread of Daoist monasteries are perhaps worthy of greater attention than the reasons underlying their origins. This point is concerned

with how we can explain the regional differences that mark the rise of Daoist monasteries, which first appeared in southeast China during the early stages of the Southern (420–589) Dynasties. Moreover, there were many more Daoist monasteries in the south than in the north for the entire duration of the Southern and Northern (386–581) Dynasties. How did this new mode of practice emerge from within the Daoism of the Southern Dynasties, and why did the majority of Daoist priests accept it so readily?

In this paper, I seek to explore the emergence of Daoist monasteries from the perspective of comparative religious history. I also attempt to highlight the role of mountain-based Daoist practice among the Celestial Masters order in the Jin and Song (420–479) periods, which led not only to the establishment of Daoist monasticism but also to a loss of purity therein.

## 2. Patterns in the Development of Monasticism

Monasticism exists in numerous religious traditions, with the most prominent examples being Christian monasteries and convents and Buddhist monasteries. These monasticisms share similar characteristics, such as disengagement from one's family in favor of collective religious practice; obedience to clear standards of behavior and religious commandments; and the possession of a unique social identity and image and pattern of social relations (Weckman 2005; Johnston 2000, p. 1; Juergensmeyer 1990). Scholars insist that Daoism is not a monastic religion because it manifests in too many non-monastic ways, especially not practicing strict celibacy (Strickmann 1978; Schipper 1984). Yet, in the Tang dynasty (618–907), entering the monastic life became the only legal way for Daoist priests to practice, as prescribed by official law and religious code (Zheng 2004). Thus, more than a hundred years after the appearance of the earliest Daoist monastery, Daoism in the Tang Dynasty had already established monasticism, and although it later underwent many new changes, to this day, the largest sect of Chinese Daoism, *Quanzhendao* 全真道, still adheres to the tradition of monasticism.

Understanding monasticism as it developed in other religions will be of some use to our understanding of the emergence of the Daoist monastery (Kohn 2003). Christian and Buddhist monasticism both originated with cloistered monks who followed hermitic and ascetic practices. Christian monasticism stems from the Desert Fathers of fourth-century Egypt, led most notably by St. Anthony of Egypt (c.251–356), who forsook an affluent home life for a solitary, ascetic life in the wilderness over a period of several decades. Little by little, his admirers grew in number, and modeled their behavior on his own; St. Anthony then led these ascetics, guiding them together in the monastic life (Harmless 2004). Similarly, Buddhist monasticism emerged from hermitic and ascetic Indian traditions. The path taken by Sakyamuni himself bore some resemblance to that taken by St. Anthony: he abandoned a life of material ease, renouncing his family to become an ascetic. He pursued religious practice in mountains and forests, and even among tombs, and he attracted a band of followers. During the Buddha's time, *sangha* (groups of monks) still dwelt primarily in forests and begged for food as they wandered the streets. By the time of the Mauryan Empire (ca. 324–187 BCE), more and more Buddhist monks stopped wandering and started to gather next to pagodas where relics were enshrined, thereby establishing permanent, organized *samghārama* with a large complex of buildings. From "communities of wanderers", Buddhist monks evolved into monastic groups with fixed dwellings (Lamotte 1988, pp. 310–13; Keisho 1980, pp. 287–322).

Christian and Buddhist monasticism developed along similar trajectories. They both witnessed an aggregation of monastic sites, a move toward fixed dwellings among ascetic groups, and the institutionalization of the ascetic life. First, a few ascetics, having actively cast aside the secular life, established simple, non-permanent dwellings away from centers of human activity. Soon thereafter, others began to follow in their footsteps and assemble in such locations, leading to the formation of ascetic groups. Then, this group coalesced even more and began to build fixed premises. Ultimately, having consolidated their dwellings and laid down their commandments, a monastic system emerged that was based on ascetic communal living.

There are, moreover, certain universal features that gave rise to monasticism. Monastic movements are often backed by specific political and economic contexts, especially social crises and ideological changes. The general spiritual anxiety that social crises create is sufficient to generate widespread avoidance behavior and provides such behavior with room to thrive. Furthermore, monasticism is a radical religious undertaking; it is a form of resistance and innovation in the face of old ascetic methods that have lost their spiritual and charismatic appeal. Buddhism is considered to have been a reaction against ritual sacrifice and nature-worshipping Brahminism. Fourth-century Christian imperialization, together with the corrupt practices that engendered, is considered to have been an important factor in the rise of monasteries (Dunn 2003, pp. 1–2). In the words of James W. Thompson, "[m]onasticism, with its other-worldliness, its self-abnegation, its austerity, was a protest against the worldliness, the riches, the vanity of a church grown scandalously corrupt." (Thompson 1928, p. 138).

It is ironic that the establishment of monasticism was frequently a betrayal of its initial purpose. Collective monastic life developed out of hermitic and ascetic practices, characterized by a withdrawal from secular life in organized society. Yet, despite this, once it became organized and its locations became fixed, it inevitably brought a certain formalization and dogmatization to religious life, leading to the forging of new social relations; that is, it changed secular life. Such a change comes close to Max Weber's notion of the "routinization of charisma". This quotidianism and routinization was undoubtedly contrary to the original intention behind monasticism, and frequently inspired new waves of rebuilding of the monastic tradition. For this reason, monasticism is often represented in religious history as a cycle of corruption and reform. Christianity's so-called "desert city paradox" and Mahayana Buddhism's Forest Hypothesis are both relevant in this regard (Silber 1995, p. 137; Drewes 2018).

### 3. Daoist Practice in the Mountains

It may be said that monasticism itself is an amalgam of two rather contradictory tendencies: on the one hand, it has an anti-communitarian element with an emphasis on an asceticism that renounces secular life. On the other hand, it also has a communitarian element, with groups of ascetics residing communally in monasteries. From a collective standpoint, it sets itself apart, founding a new community while distancing itself from society at large. In terms of early Daoism, these two qualities belong to two different traditions: one is the first organized Daoist movement that emerged in the second century, the Celestial Masters, and the other is the long-standing Daoist hermetic tradition, dedicated to the pursuit of transcendence through individualized cultivation, often referred to in the texts as *xiandao* 仙道, or "the way of transcendence" (Campany 2009). Chinese researchers often refer to the latter as *shenxian daojiao* 神仙道教 or "The Taoism of the Immortals" (Hu 1989). In this paper, I will simply refer to this as the hermetic Daoism.

The Celestial Masters' organization relied on and served village-level civil society, preaching to all people. They were not concerned with seclusion or ascetic practices. On the contrary, they were highly attentive to family values and daily life (Stein 1979). In the late Han dynasty and in the Wei and Jin periods, the Celestial Masters gained widespread popularity, forming a relatively large-scale religious order. Although its history was even longer, hermetic Daoism was not devoted to ecumenical missionary work, and was only concerned with individual asceticism or esoteric knowledge transmitted from master to disciple (Tadao 1984, pp. 425–61). Since monasticism began with those who practiced seclusion and asceticism, we should begin our investigation of Daoist monasticism by examining hermetic Daoism.

Analogous to the Desert Fathers of Christianity and the "Forest Saints" of Buddhism, the ascetics of ancient China were usually referred to as "Men of the Cliffs and Caves" (*Yanxue zhishi* 岩穴之士) (Vervoorn 1990). It was said, furthermore, that "to practice Daoism you must enter the wooded mountains 為道者必入山林." (*Baopuzi neipian Jiaoshi* 抱朴子內篇校釋, p. 187; Michael 2016) Mountains are to Daoism as the desert is to Christianity, and as forests

are to Buddhism; they are symbolic of another world. However, the early Daoist monastic tradition embodied a completely different set of beliefs and values to those of Christianity and Buddhism.

Wolfgang Bauer stated in this regard that "the degree of rigidity to which eremitism was pursued can be connected with that basic concept of the universe: the less pessimistic the latter was, the less radical the intended degree of withdrawal from the world." (Bauer 1981). Hermetic Daoism altogether lacks conceptions of the suffering of life or the sinfulness of life; rather, it considers that being born human is a blessing; therefore, one needs to maintain one's health and live a long life. Hermetic Daoism even tends toward hedonism. Similarly, it differs from the Christian and Buddhist emphases on ascetic practice in the desert and forest, respectively, and on the significance of trial by fire. In hermetic Daoism, mountains are considered a "happy place" (*fudi* 福地) where one can maintain one's health and even avoid misfortune. When ascetics practiced in the mountains and engaged in such related practices as abstinence from grains and from physical desire, their aim was not self-redemption or purification of the mind, but rather themes to which Daoism has always aspired: longevity and immortality, that is, the immortality and transformation of the body (Eskildsen 1998). *Daoji jing* 道機經, the Daoist ascetic manual that gained widespread currency during the Wei and Jin periods, notes that "there are too many desires in this world 民間多慾" and that there is a need to "go into the mountains and live in seclusion, to not stray from naturalness, and to keep practicing Daoism to the point where one is transformed; then, one can gain immortality 入山潛處, 守志自然,功滿形變, 則得長生." The Daoji scriptures also speak in detail of health, diet, and carnal knowledge. It is thus clear that "going into the mountains" did not mean having to follow ascetic practices (*Xiandao jing* 顯道經, p. 646; Sun 2013). Wang Zhen 王真, a Daoist priest during the Cao Wei period, brought three concubines into the mountains with him as part of his spiritual practice. This was nothing out of the ordinary for Daoists (*Shenxian zhuan jiaoshi* 神仙傳校釋, p. 218). Thus, although the practice of hermetic Daoism in the mountains tended toward seclusion, it lacked an ascetic tradition. Rather, it was a "mildly" monastic tradition that embraced seclusion but not asceticism.

Following the political crises and social turmoil of the Eastern Han dynasty (25–220), withdrawal from society to live in seclusion became a notable social movement at the end of the Han dynasty and into the Wei and Jin periods, a clear indicator being the official *Hou Han shu* 後漢書, which for the first time, included biographical accounts of hermits. It was also during this period that the pursuit of longevity became widespread across all levels of society. In the *Shenxian zhuan* 神仙傳 by Ge Hong 葛洪, generally all of the people who became "immortal" had undertaken Daoist practice in the mountains. Moreover, in secular texts, we also read of many ordinary people who turned to asceticism in the mountains. The *Shuijing zhu* 水經注, published in book form in the early sixth century, refers to the traces left behind by numerous ascetics. It can be seen that the pursuit of Daoism in the mountains was, at the time, a widespread phenomenon across northern and southern China, one difference being that northern ascetics largely lived in stone chambers, while their southern counterparts mostly constructed temple buildings. Most of their cloisters, located as they were in wooded mountains, were relatively crude, and many were soon abandoned, which highlights the lack of stability of these environments for the individual practice of Daoism (*Shuijing zhu jiaozheng*, pp. 44, 104, 225, 650, 660, 715, 750, 753, 796).

The records in the *Shuijing zhu* indicate that mountain-dwelling ascetics from the period lived in communities, as opposed to living as individual hermits. For instance, we read that on Xiyi Mountain 錫義山 "there are currently dozens of Daoist priests in attendance, with their hair hanging down loosely, using atractylodes for food 今有道士被髮餌朮, 恒數十人"; in the Wudang Mountains 武當山, "there is a gathering of people taking herbal medicine for their health 藥食延年者萃焉"; and on Qingxi Mountain 青溪山, "besides the spring waters, there are many cabins built by Daoist priests for their spiritual practice 泉側多結道士精廬." (*Shuijing Zhu jiaozheng*, pp. 660, 753) Similarly, the *Hou Han shu* observes that Liu Gen "lives in seclusion on Song Mountain 嵩山; many people come from distant

places in the hope of learning Daoism from him 隱居嵩山中, 諸好事者自遠而至, 就根學道"
(*Hou Han shu*, p. 2746); in the *Guanzhong ji* 關中記, Pan Yue 潘嶽 remarks that on Song
Mountain there are "more than ten caves 石室十餘孔" and that "many Daoist priests dwell
inside, to distance themselves from the secular world 道士多游之, 可以避世" (*Chuxueji*,
p. 103); the *Nan Yongzhou ji* 南雍州記 describes how on Qingxi Mountain, "hermits come
and go, and there are often more than one hundred of them 學道者常百數, 相繼不絕."
([Huang 2000](#), p. 568) Gatherings, such as that described in the *Hou Han shu*, were held
on an even larger scale: a hermit named Zhang Kai 張楷, for instance, "lives in seclusion
on Hongnong Mountain. His students follow his lead, and the place in which they live
has become a marketplace 隱居弘農山中, 學者隨之, 所居成市." (*Hou Han shu*, pp. 1242–43)
The *Jin shu* 晉書 remarks that Guo Yu 郭瑀 "is a hermit in Linsong Xiegu, and has dug out
a cave in which to live 隱於臨松薤穀, 鑿石窟而居" and that "he has over one thousand
registered students 弟子著錄千餘人." (*Jin shu*, p. 2454) The "Shilao zhi 釋老志" of *Wei shu*
魏書 observes that during the Sixteen Kingdoms period (304–439), alchemist Lu Qi 魯祈
"lived in seclusion on Han Mountain and educated several hundred students 避地寒山,
教授弟子數百人." (*Wei shu*, p. 3054) Gathering in the mountains was rarely witnessed
before this period and is related to the loosening of the state's grip on the lives of ordinary
people from the late Eastern Han period.

It is noteworthy that even though quite large groups of people who had withdrawn
from society did gather in mountain regions, hermits in the Wei and Jin periods failed to
develop fixed temple buildings. The *Shuijing zhu* records the presence of a large number
of Buddhist temples and folk ancestral temples in northern China but does not identify the
existence of a single Daoist monastery ([Chen 1985](#), pp. 252–61; [Hisayuki 1964](#), pp. 366–90;
[Cai 2011](#), pp. 130–41). There were even places of communal Daoist worship adjacent to
Buddhist temples with no sign that they ever developed into Daoist monasteries. The
classic example of this can be seen from the Daoist ascetics Wang Jia 王嘉 and Zhang Zhong
張忠 who lived in northern China (controlled by northern ethnicities) following the Revolt
of Yongjia 永嘉之亂 that occurred in 311 CE. The *Jinshu* records the following:

> [Wang Jia] didn't partake of the five cereals, nor did he wear fancy clothes. He
> practiced the art of breath straining, and didn't make friends with worldly people.
> He lived in seclusion in Dongyang valley, digging out a cave on a cliff as a place
> to live. He had many hundreds of followers, and they also lived in caves. In
> the final years of Shi Jilong's reign (Shi Le, pp. 295–349), Wang Jia abandoned
> his disciples, went to Chang'an, lived in seclusion in the Zhongnan Mountains,
> building a thatched hut as his residence. When his disciples learned of this, they
> once again came in search of him. Wang Jia then dwelt in solitude on Daoshou
> Mountain. (*Jin shu*, p. 2496)

> [王嘉]不食五穀, 不衣美麗, 清虛服氣, 不與世人交遊。隱于東陽穀, 鑿崖穴居,
> 弟子受業者數百人, 亦皆穴處。石季龍之末, 棄其徒眾, 至長安, 潛隱於終南山,
> 結庵廬而止。門人聞而復隨之, 乃遷於倒獸山。

The path taken by Wang Jia strongly resembles that taken by St. Anthony. To prevent
spiritual regression, the latter parted ways with his group once he was surrounded by
people in his cloisters and headed to an even more remote place to continue his practice.
This gradual retreat from society is extremely similar to the two instances in which Wang
Jia "abandoned his disciples".

The example of Zhang Zhong, by contrast, reveals not only that mountain-dwelling
hermits of the era formed monastic social groups of a certain size. It also highlights the
practice of Daoism in groups under the guidance of a teacher.

> During the Revolt of Yongjia, Zhang Zhong lived in seclusion on Tai Mountain.
> His mind was still, and without desire. He practiced breathing techniques, took
> fungus and minerals as medicine, and practiced ways to maintain his health….He
> lived in a quiet and secluded valley between towering cliffs. He had dug out
> a cave as his room. His students also lived in caves, at a distance of more than

60 steps. Every five days, they would come and pay their respects to him. Zhang Zhong educated his students not by speaking but rather through his behavior. The students studied his behavior by observing him, and then withdrew. Zhang Zhong built a Daoist altar in the cave, where he worshiped every day. (*Jin shu*, p. 2451)

[張忠]永嘉之亂，隱于泰山。恬靜寡欲，清虛服氣，餐芝餌石，修導養之法……其居依崇岩幽谷，鑿地為窟室。弟子亦以窟居，去忠六十餘步，五日一朝。其教以形不以言，弟子受業，觀形而退。立道壇於窟上，每旦朝拜之。

This appears to have been the highest form of monasticism among fourth-century northern ascetics. According to the *Shuijing zhu* and the *Gaoseng Zhuan* 高僧傳, Zhang Zhong's ascetic group happened to be in the same place at the same time as the monk Zhu Senglang's 竺僧朗 Buddhist order, yet Zhang simply "stayed in his cave 穴居". Senglang then "built a formal temple and houses, with the buildings joined together 大起殿舍, 連樓累閣". "The complex had several dozen rooms, and it is said that after this was done, more than one hundred followers visited 內外屋宇數十餘區, 聞風而造者百有餘人." (*Shuijingzhu jiaozheng*, p. 209; *Gaosengzhuan*, p. 190) Such a comparison reveals that the Buddhism practiced in the mountain forests was already a fairly complete monastic system, while the mountain-dwelling Daoist ascetics represented by Zhang Zhong had yet to take such a step toward developing a system of their own.

The phenomena noted above—of entering mountain areas for religious practice, and the clustering of groups who did so—primarily emerged in northern China. Considering monasticism's general pattern of evolution, the potential for Daoist temples to evolve was far greater in northern China. Nonetheless, it was in the southeast during the early Southern Dynasties period that the Daoist monastic movement initially unfolded. Moreover, Daoist monasteries in southern China far outnumbered those in the north for the entire duration of the Southern and Northern Dynasties period. This disparity differed greatly from the parallel development of fourth-century Buddhist temples across northern and southern China. Wei Bin 魏斌 considers this glaring regional disparity to be due possibly to the general mood of mountain-based religious practice shifting south into the Jiangnan region in the wake of the Disaster of Yongjia. It was in Jiangnan that Daoist monasteries developed and grew in sophistication, while in the north they gradually disappeared (Wei 2017, pp. 129–30).

The seclusive atmosphere in the Jiangnan region during the Six Dynasties period (222–589) indeed surpassed that of the north, and it also shaped the common social practices of the seclusive scholar–officials. Correspondingly, high officials and the nobility became accustomed to "recruiting hermits" (*zhaoyin* 招隱) and were happy to establish premises for them (*Shishuo xinyu jianshu* 世說新語箋疏, p. 778; *Song shu* 宋書, pp. 2276–77, 2291). In this sense, early Daoist monasteries were indeed a manifestation of Southern seclusive culture. The earliest Daoist monastery, Chongxu Guan, constructed in the third year of the Taishi 泰始 reign (467), may be regarded as an "academic hall" (*xueguan* 學館) established by the state (Akiko 2009, pp. 234–37). Moreover, several other Daoist monasteries sponsored privately by aristocrats may be regarded as having resulted from the same logic; these were merely a form of private sponsorship.

However, understanding the emergence of these monasteries as a version of Daoism that sought sponsorship still poses problems. First, there are even earlier instances of recruiting Daoist priests to establish premises, such as Cao Cao's 曹操 establishment of humble thatched-roof abodes (*maoci* 茅茨) for Xi Mengjie 郄孟節 (*Shenxian zhuan jiaoshi*, p. 218); the Northern Wei (386–534) emperor Daowu's 道武帝 creation of a "hall of quietude" (*jingtang* 靜堂) for the immortal court academician Zhang Yao 張曜; and the establishment of a hall for Du Zigong 杜子恭 by the Eastern Jin's (317–420) Huan Wen 桓溫 (*Taipingyulan*, p. 765). Yet none of these buildings saw the advent of an "era of the Daoist monastery." Furthermore, what appears in the records are primarily the most famous Daoist monasteries of the Southern Dynasties period, whose heads were relatively "successful" Daoist priests from the Southern Dynasties' social class of scholar–officials (Pettit 2013). The speed with which the Daoist monasteries spread in southern China proves the existence of a broad social

base for Daoist practice. In other words, the Daoist community in the South was ready for a shift in the direction of Daoist monasticism. This was not something that individual religious leaders could influence, nor was it something that official sponsorship could restrict. We should now turn our attention toward the religious choices made by southern China's grassroots Daoist believers, who were primarily priests in the Celestial Masters.

### 4. The Celestial Masters' Entry into the Mountains

The Wei and Jin periods were a time in which the Celestial Masters disseminated widely across northern and southern China. Yet, as a result of constraints at the heart of the religion, the spread of the Celestial Masters across both the north and south primarily took the form of the spontaneous duplication of a libationary system (*jijiu tizhi* 祭酒體制). This led to a situation in which "everyone referred to themselves as a teacher and established their own parish 人人稱教，各作一治". The original organization, the rites, and even the doctrine had spiraled out of control. The scattered, self-supporting parishes of the Celestial Masters had become virtual money-making tools of the libationers (*jijiu* 祭酒), to the extent that there was mutual competition and factional fighting. This chaos was vividly depicted by Lu Xiujing 陸修靜 in the *Daomen kelüe* 道門科略 and in contemporaneous scriptures. The irregular development of the Celestial Masters gave rise to status differentiation among the Celestial Master libationers. In southern China, there were large parishes overseeing tens of thousands of believers, like Du's parish in Qiantang 錢塘, and mid-size orders with over 800 believing households; there were also "lesser masters" (*xiaoshi* 小師) who wandered the streets, being unable as they were to establish parishes. The status of libationers varied from person to person. Many folk-based Celestial Master libationers had no means of effectively overseeing their parishioners; all they could do was wander, performing the low-status role of the professional religious service provider. Having entered the fifth century, the Celestial Masters were confronted by the impact of Buddhism and suppression by secular regimes. In particular, the Sun En 孫恩 Revolt of 399–411 threw the Celestial Masters order, which had pursued a policy of "mastering the households and ruling the people 領戶治民", into an unsustainable position (Sun 2020).

It was around the year 400 that the Shangqing scriptures 上清經 and the Lingbao scriptures 靈寶經, which had newly emerged in southern China, began to disseminate widely. The popularity of these scriptures, which were later referred to as the "Sandong Jingshu 三洞經書", is not unrelated to the lack of sanctity ascribed to the Celestial Masters order. These scriptures advocated anew the hermetic tradition of longevity and immortality while selectively absorbing elements from Buddhism to form a scriptural system with a new cosmology and outlook on the world. The texts were styled as "the Supreme Way" (*shangdao* 上道), the "Supreme Scripture" (*shangjing* 上經), and the "Supreme Law" (*shangfa* 上法), and were heavily critical of old Celestial Masters' amendments to Daoist law, which were disparaged as "the lower way" (*xiadao* 下道) and "the lesser way" (*xiaodao* 小道). The primary audience comprised ascetics who sought to break away from secular society and pursue holiness; this naturally included Celestial Master believers.

The Shangqing scriptures are regarded as the pinnacle text of hermetic Daoism principles and practice in pursuit of individual longevity and immortality (Strickmann 1979; Robinet 1997). The earliest members of the Shangqing school were Xu Mi 許謐 and his son–originally members of the Celestial Masters–from Jurong 句容, as well as the medium Yang Xi 楊羲. The Shangqing scriptures inherited hermetic Daoism's belief in mountain-based spiritual practice and held that practitioners should "relinquish their families, divorce their wives, go to the Five Great Mountains and engage in prolonged fasting in the mountain forests 棄家放妻, 游五嶽, 長齋山林." (*Sijimingke jing*, p. 426) In the *Zhen'gao* 真誥, it states that Yang Xi worked tirelessly to convince Xu Mi to build a temple building on Mao Mountain, to complete the transformation from the "Xu chief of staff in the human world 人間許長史" to the "Daoist priest Xu in mountains 山中許道士." Xu Mi's temple building was also known as "Nanshan zhi 南山治." It could be said that this was the earliest case of the Celestial Masters moving into chambers of quietude in

the mountains. However, the Shangqing Daoism was still an individualized religion, and Xu Mi's chambers of quietude on Mao Mountain were only spaces for individual practice and soon ceased to exist. They cannot be viewed as a Daoist monastic complex as such (Pettit 2013, pp. 14–49).

Of greater revolutionary significance was the emergence of the Lingbao scriptures, which were essentially a result of the development of hermetic Daoism. They emphasized mountain-based practice but accepted and transformed a large number of Buddhist beliefs and values. The Lingbao scriptures are considered the first Daoist scriptural system to have been truly pervaded by Buddhism, or to have attempted to hyper-assimilate Buddhism. Among the most important of these beliefs and values are universal salvation, merit transfer, and asceticism (Zürcher 1980; Bokenkamp 1983). Through these concepts, the Lingbao scriptures redefined the responsibilities and the image of the Daoist priesthood: priestly asceticism was not merely concerned with the pursuit of immortality; it was even more important to help liberate all sentient beings. Likewise, liberation from personal desire did not require first-hand study of Daoism; by inviting a Daoist priest to make a sermon or perform a ritual, one could return merit to oneself. Merit transfer required the assurance of sanctity; as a result, a professional Daoist priesthood emerged, with members who had broken away from the secular world. To ensure the sanctity of the Daoist priesthood, the Lingbao scriptures emphasized the importance of mountain-based ascetic practice; furthermore, in order to pray for the salvation of common people, it was necessary for professional Daoist priests to hold collective Lingbao abstinence rituals in "abstinence halls" (Campany 2015). Yet, as Stephen Bokenkamp has noted, "while the Lingbao authors seem to have imagined no dedicated structures for their practice beyond expanded versions of the already extant Chambers of Quietude, they did expect their monastics to be dedicated religious specialists—anchorites, rather than cenobites." The emergence of Daoist monastery was "not fully prefigured in the Lingbao scriptures." (Bokenkamp 2011, p. 124).

Newly emerged southern Daoist scriptures, particularly the Lingbao scriptures and their new beliefs and values, not only provided Daoist believers with a path to redemption that was new and original, rarefied and sacred, but the modes of practice they advocated also corresponded with the contemporaneous and actual needs of the crisis-ridden Celestial Masters order—and in particular, the needs of the grassroots libationers. The Celestial Masters readily accepted the "Sandong Jingshu" once it emerged; this served to deepen the transformation of the Celestial Masters' doctrinal beliefs and values, practices, and the organizational model of the order. Yet this process of acceptance was complex and was a strong reflection of the active trade-offs, adjustments, and transformations undertaken by the order. Contemporaneously, a number of "hybrid scriptures" appeared, ones in which the Celestial Masters were considered the standard and in which the doctrines and rituals of the Celestial Masters, the Shangqing scriptures, and the Lingbao scriptures were blended. These were most likely authored by members of the Celestial Masters. With regard to the acceptance and fusing of the newly emerged doctrine, the methods and points of emphasis within these scriptures were wholly dissimilar, which demonstrates something of the complexity of the various stages within the evolution of the Celestial Masters order.

Appearing in book form in the Jin-Song transitional period, the eschatological *Dongyuan Shenzhou Scriptures* 洞淵神咒經 were a relatively early manifestation of the "hybrid scriptures". These scriptures are thought to have been composed by Daoist priests from a lower socio-economic class of Jiangnan society; they are a patchwork of popular religious elements seeking to promote salvation in the last days, and the targets of their proselytizing were, to a significant extent, grassroots Celestial Master believers (Lü 2008, pp. 174–81). By means of radical proselytization, the *Dongyuan Shenzhou Scriptures* sought to popularize a new form of spiritual practice among Celestial Master believers and to greatly esteem "the severing of all ties, and the practice of the Dao in the mountains 一切斷絕, 入山修道", while also permitting "being among the people, and treating their illnesses 遊行世間, 為人治病." (*Dongyuan Shenzhoujing*, p. 34) The *Dongyuan Shenzhou Scriptures* do not entirely discount

the value of being among the people as traditionally practiced by priests of the Celestial Masters order. It even goes so far as to promise that they, just like the Daoist priests of the mountain forests who practiced the asceticism of the "Sandong Jingshu", would not suffer from pestilence. However, it also holds that only the latter group, the mountain-dwelling priests of the "Sandong Jingshu", were true Daoist priests and that others were merely "minor priests 小小道士." (*Dongyuan Shenzhoujing*, p. 78) It also constantly emphasizes that it was only Daoist priests in the mountains who would be able to endure future disasters. Volume two of the *Dongyuan Shenzhou Scriptures* envisions the mountain-based spiritual practice of the Celestial Masters order as follows:

> From this the Renwu year onward, Daoist priests should wear religious garments such as hats, coarse cloth, aprons and capes, and carry a staff. They eat only one meal a day, and not consume food after midday. They stop eating all meat and drinking alcohol. They educate people from secular society, and do not violate any laws. Men and women teach each other, and be mentored by the wise. They cannot raise living things, and live alone. It is their duty to study the scriptures. People living in the mountains should abstain from meat and fish every month; during this period of abstinence, they may only eat vegetarian food. Whether there are ten, thirty or one hundred people living together in the mountains, they should cultivate a large area of fields and gardens, and plant trees and vegetables; they should build quadrangular houses as well as an abstinence hall and pavilion. One, two or three people should not live separately from others, because otherwise the spirits will deceive them; the ways of practice within the scriptures cannot be undertaken on one's own. The abstinence hall may hold many people. People must fast three times a month, burn incense three times a day, and pray in ten directions: then the gods will attach themselves to people. What difference does it make if one or two people are alone in the mountains, not fasting according to the prescribed methods and just living there on their own with nothing more than insects and deer around them? They should follow wise teachers, those with many skills and scriptures. There should be more than one person present; the more there are, the better. There is no need to stop at a dozen or more people, much less two or five. Having just a few people is not enough to subdue the mountain spirits. Mountain spirits are deceitful, and that means bad luck. (*Dongyuan Shenzhoujing*, p. 78)

> 道士自今壬午年以去，亦作冠褐裙帔三法衣策杖耳。一日一食，過中不餐，斷一切葷，酒亦不嘗。教化俗人，為事不得犯科。男女相度，智者為師。不得畜生生之物，正孑然一身耳。經書為業，入山中人，月月長齋，齋空食菜耳。入山十人、三十人、百人一處，廣作田植、園菜、五果、屋舍四方，並齋堂、樓閣，不得一人、二人、三人獨住。鬼神欺人，經法不可獨。齋堂人多，月月三齋，日日三時，上香禮拜十方，此為神來附人矣。二人一人獨在山中，複不立法建齋，直獨在山中，蟲鹿亦在山中耳，此為何異也。當奉明師，師多才技，又多經文，乃一人上，可住多多。遂上十人者，不可住也，況複二人、五人也。此不伏山神，山神欺人，人亦不吉矣。

This is an extremely important text in the context of the establishment of Daoist monasticism. It reveals that amid the stimulus provided by the emergence of new scriptures, the Celestial Masters order had already laid the ground rules for monastic practice: priests dwelling in the mountains should form groups of more than ten people; to live collectively in a fully equipped temple, farmland and vegetable patches need to be established as a source of income; to undertake collective spiritual practice in a timely manner; and to abide by relevant commandments and systems. Clearly, this closely resembles a genuine monastic lifestyle. Yet within these monastic groups, there was still no explicit limitation on gender roles, only generalized claims that "men and women instruct each other, and are mentored by the wise", giving the impression that the organization's living space comprised a mix of males and females.

The mountain-based asceticism advocated in the *Dongyuan Shenzhou Scriptures* differed from what was intended in the Shangqing scriptures and the Lingbao scriptures. The *Dongyuan Shenzhou Scriptures'* most striking characteristic was their distinctive collegiality. The mountain-based practice they sought was not characterized by seclusion and asceticism, but more closely resembled a revamping of an existing form of Celestial Master activity. The scriptures did not relinquish the secular flavor of the original Celestial Masters; while even stronger in their praise of mountain-based practice, they still pursued "the use of education to enlighten people 人間教化" such that there was no distinction between being in the mountains and being in the world. The scriptures even asserted that "if Daoist priests remain in the mountains, where the roads are far-flung, people in secular society cannot find them, and they have no way of converting. Even though they may wish to, there would be nowhere to receive instruction. Therefore, wise Daoist priests do not necessarily reside in the mountains 道士入山, 山途玄隔, 世人不見, 無處歸依, 雖有本心, 無處相度, 是以智人道士, 不必山中矣." (*Dongyuan Shenzhoujing*, p. 29) We may well say that as a monastic ideal, this was less than complete, yet it highlights the Celestial Masters' experimentation with, and transformation of, new religious beliefs and values in the Jin and Song periods. It is precisely because of this adaptability that the Celestial Masters were able to attract such broad-based, grassroots support and engagement from their followers in southern China, thus acting as a catalyst for the spread of the Daoist monasticism in southern regions.

## 5. Southern Dynasties Daoist Monasteries and the Celestial Masters

The earliest Daoist monasteries emerged in the south in the roughly 50-year period after the publication of the *Dongyuan Shenzhou Scriptures*, or at least they appeared in historical records as such. These monasteries were all essentially built via imperial edict or through funds donated by scholar–bureaucrats. Most of the heads of the monasteries were not normal Daoist priests at the grassroots level, but higher-ranking and intellectually minded Daoist priests with close connections to the social class of scholar–officials. For this reason, scholars frequently emphasize the proximity between early Daoist monasteries and the chambers of quietude. However, this view has to a significant degree been distorted by the retention of historical data: these well-known Daoist monasteries were merely a drop in the ocean of the Daoist monasticism, and a great many monasteries at the grassroots level never entered the historical record.

One item to which we can refer is the *Jiuxi Zhenren Sanmao Jun Stele* 九錫真人三茅君碑 erected at Mao Mountain in 522 CE. The inscription on the reverse side of this stele lists the names of 103 Daoist priests and 63 Daoist monasteries or temple buildings; very few of these are to be seen in other records (Sun 2014, pp. 99–100). Moreover, these monasteries' names were merely those from the vicinity of Mao Mountain during the Southern Liang dynasty (502–557), and their number exceeds the sum total of the Southern Dynasties' Daoist monasteries recorded in other textual sources. Where did all these Daoist monasteries spring from? In his notes on his depiction, contained in the *Zhen'gao*, of the state of ascetic practice between the early Liu Song and late Qi (479–502) periods in the vicinity of the southern caves of Mao Mountain, Tao Hongjing 陶弘景 offers an important clue:

> Currently, around the entrances to the great southern cave of Mao Mountain, there is a good water source but many rocks, and it is flat down below. In the early years of the Liu Song period, only the female Daoist priest Xu Piaonu lived here. She obtained funding from the Guangzhou governor Lu Hui. She lived at the entrance to the cave, and passed away after living here for several years. [Her] disciple, surnamed Song, was a very noble woman who was undisturbed by the outside world. She died at an old age, and was buried on the southern side of the mountain. Song's disciple, surnamed Pan, continued to live here and is still alive today. During the Yuan Hui reign (473–477), some men also came to live alongside them on the southern side. During the early Southern Qi dynasty, [the emperor] ordered Wang Wenqing, who was from Jurong, to

establish a Daoist monastery here and name it "Chong Yuan". A temple building and corridors were built in a very imposing style. There are seven or eight Daoist priests, and they all receive official funding. For more than twenty years now, men and women from near and far have been meeting miles around, and have established more than ten buildings. But very few of them practice "the Supreme Way"; most of them practice the Lingbao rites and talismans. Recently a woman came to live at the entrance to the cave; she often cleans and sweeps and Claims to be the administrator of the cave. She often practices divination like a sorceress, and is pretentious and ostentatious. Situations like that are everywhere. There are also streams on the eastern and western sides of Mao Mountain, and there is also the spot where Ren Dun, who attained the Dao, lived in the waning years of the Jin dynasty; the stove he used to refine his potions is still there today. Now, Xue Biaozhi and others live there. There is also Zhu Fayong, who lives on a small hill nearby; it boasts a fine view, but it lacks a water source. (*Zhen'gao*, p. 558)

今[茅山]近南大洞口有好流水而多石，小出下便平。比世有來居之者，唯宋初有女道士徐漂女，為廣州刺史陸徽所供養，在洞口前住，積年亡。女弟子姓宋，為人高潔，物莫能干，年老而亡，仍葬山南。宋女弟子姓潘，又襲住，於今尚在。元徽中，有數男人復來其前而居。至齊初，乃敕句容人王文清仍此立館，號為崇元。開置堂宇廟廊，殊為方副。常有七八道士，皆資俸力。自二十許年，遠近男女，互來依約，周流數里，廟舍十餘坊。而學上道者甚寡，不過修靈寶齋及章符而已。近有一女人來洞口住，勤於灑掃，自稱洞吏，頗作巫師占卜，多雜浮假，此例亦處處有之。大茅東西亦有澗水，有晉末得道者任敦住處，合藥灶壚猶存。今有薛彪數人居之，又有朱法永，近小山上，快矖眺而乏水。

Through this record, we can see that the Mao Mountain Daoist monastery emerged in the early years of the Liu Song period. When it first began, private monastery buildings for Daoist practice were built in a piecemeal fashion by mountain-dwelling ascetics. Afterward, officials of the early Southern Qi period founded the "Chongyuan Guan 崇元館", which, compared with the earlier monastery buildings, was much more spacious and formal. Then, more ascetics assembled near monastery buildings, establishing more than ten "public offices" (*xieshe* 廨舍), which were less formal. These monastery buildings, built by mountain-dwelling ascetics who gathered spontaneously, were not established by official or imperial order whatsoever, but there is evidence to suggest that they were referred to as Daoist monasteries or subsequently evolved into such (Bumbacher 2000, pp. 442–43). For instance, Zhu Fayong 朱法永, living on his small hill, was most likely the very same "Zhu Fayong of Yanguan, owner of Dongxuan Guan 洞玄館主鹽官朱法永" mentioned on the reverse side of the *Jiuxi Zhenren Sanmao Jun Stele*. From this we can infer that the large number of Daoist monasteries' names recorded on the reverse side of the stele, such as Long'e Guan 龍阿館, Fuxiang Guan 福鄉館, Jinling Guan 金陵館, Fangyu Guan 方隅館, Tianshi Guan 天市館, Beidong Guan 北洞館 and Maozhen Guan 茅真館, all bear the clear imprint of Mao Mountain's sacred geography, which would have made them similar to the Daoist monastery built by the ascetics on Mao Mountain.

There is evidence to suggest that many of these mountain-dwelling ascetics were previously priests of the Celestial Masters order. For instance, cousins of Tao Hongjing's disciple Zhou Ziliang 周子良 came to Mao Mountain from eastern Zhejiang in the twelfth year of the Tianjian reign (513) and lived in "annexed buldings on the western hill" (*xi'e biexie* 西阿別廨) (*Zhoushi mingtongji* 周氏冥通記, p. 158). Zhou Ziliang's maternal aunt Xu Baoguang 徐寶光 was originally "a libationer of the obsolete Daoism" (*jiudao jijiu* 舊道祭酒). Before coming to Mao Mountain, she "left home at the age of ten, studied Daoism with a teacher, set up a temple building in Yuyao 十歲便出家, 隨師學道, 在余姚立精舍", then lived in a "hall of parishes of the Celestial Masters" (*tianshi zhitang* 天師治堂) in Yongjia 永嘉 prefecture (*Zhoushi mingtongji*, pp. 522, 533). In other words, the Xu family was a prominent family of traditional libationers of the Celestial Masters order. Yet, after Zhou Ziliang followed Tao Hongjing to Mao Mountain, the Xu clan abandoned their parish

in eastern Zhejiang 浙江; they sought shelter with Tao Hongjing, and lived in "annexed buildings" (*biexie* 別廨) in the vicinity of Huayang Guan 華陽館.

From this, we may infer that many of the "men and women from near and far" who established "public offices" in the Mao Mountain of Tao Hongjing's era were, like the Xu family, originally priests of the Celestial Masters libationary system. The reason(s) that they chose to live in the vicinity of the Daoist monastery on Mao Mountain is, to a large extent, due to the fact by that time it had already become a well-known religious center. Tao Hongjing records how during the celebration of the Sanmaojun festival each year, "officials and commoners got together. There were several hundred carriages; close to four or five thousand people; men and women, both ascetics and people from secular society. There were so many people that it was like being in a big city 公私雲集, 車有數百乘, 人將四五千, 道俗男女, 狀如都市之眾." (*Zhen'gao*, p. 557) We can infer the sheer scale of the Mao Mountain religious market. This assembly of believers "just climbed the mountain together, held the Lingbao rites, and returned once these were over 唯共登山, 作靈寶唱贊, 事訖便散"; they came primarily to hold religious services, and did not have any affiliation with a Daoist monastery similar to that of the libationers of the Celestial Masters. The official Taoist monasteries are naturally the best choice for the faithful, but even these unofficially established monasteries can get a share of the huge religious market. As Tao Hongjing remarked, among the priests of Mao Mountain of the time "very few of them practice the Supreme Way; most of them practice the Lingbao rites and talismans 學上道者 甚寡, 不過修靈寶齋及章符而已." (*Zhen'gao*, p. 558) Although the "annexed buildings on the western hill" were relatively crude, they were the same as formal Daoist monasteries in that they incorporated an abstinence hall in which the Lingbao fasting rites and the Celestial Masters' rites were held, a private room and an altar (Sun 2014, pp. 105–10). It may be seen that these Daoist priests and former libationers abandoned missionary work among ordinary people; they chose to enter the mountains and build premises there in accordance with the new beliefs and values advocated in the Lingbao scriptures, and to expand by relying on income derived from the performance of ceremonies rather than by levying tax or charging rent.

The situation at Mao Mountain was an epitome of the process of conversion to the monastic life of the southern Celestial Masters of the era. It is through this process that we can understand why many of the heads of the well-known early Daoist monasteries had connections to the Celestial Masters—many monasteries were transformed directly under the order's control. A great many elements from the Celestial Masters were preserved in monastery-based Daoist practice (Sun 2020, p. 358). To a large degree, the abandonment of traditional practices and the switch to mountain-based religious practice among grassroots Celestial Master priests is attributable to economic factors; it was not necessarily based on spiritual concerns. It is understandable, therefore, if many of the characteristics of the Celestial Masters-especially its strong ties to the family as well as to secular society-have been preserved in the ordinary Daoist monasteries of the Southern Dynasties, rather than adopting a strictly monastic approach. The *Taixiao langshu* 太霄琅書, another "hybrid scripture" from the Southern Dynasties period, describes the Celestial Masters' fusion of monastic practices (Yoshitoyo 1977; Ninji 1991).

The *Taixiao langshu* embodies both the old Celestial Masters' tradition of "mastering the households and ruling the people 領戶治民", and the more recent tradition of teaching the "Sandong Jingshu". However, Daoist priests were referred to as Daoist devotees (*daomin* 道民), and it was necessary for them to render a land tax, "Non-payment of tax means parish registers cannot be obtained 租不送者, 不得治錄"; those who learned the scriptures were referred to as "disciples" (*dizi* 弟子); and "although the latter do not pay tax, they may still be taught the scriptures 於租雖闕, 無妨受經." Although there were differences as to their garments and practice methods, Daoist devotees and disciples were both scholars who "shared a reverence for the same Daoism 同宗一道." (*Dongzhen taishang taixiao langshu*, p. 664) Teachers also taught commentaries on the *Daode Jing* 道德經 and the "Sandong Jingshu", while also holding Celestial Master memorials to the emperor and Lingbao fasting

rites. In the scriptures, ascetics can be divided into six different types: those who do not renounce the world; those who do renounce the world; those who do not renounce their family; those who do renounce their family; those who wander; and those who live in seclusion. However, there was no imposition on them to enter the mountains or renounce their ways (*Dongzhen taishang taixiao langshu*, p. 668). As with the *Dongyuan Shenzhou Scriptures*, the *Taixiao langshu* held that careful attention to detail was the most important consideration, to deliver common people from torment, and that "there is no difference between living in the mountains and in the world 居山, 處世無異." Thus, it opposed the purely hermetic Daoist-style practice whereby individuals went into the mountains in pursuit of self-liberation, and promoted the establishment of premises in which to preach to ordinary people:

> People of the "Greater Vehicle" who study the Supreme Way, practice Daoism and enlighten ordinary people do not need to escape into the mountain forests. There is no social contact in the mountains. There is an abstinence from desire, the preparation of herbal remedies, and the learning of alchemy, (but) these are just the minutiae of building merit, not the most important foundation. There is no way to build merit in the mountains, so one needs to enter the big, worldly cities and build Celestial Master halls of parishes or monasteries, copy the scriptures, proclaim the wonders of Daoism, assist the state in providing relief to the common people, be diligent in one's spiritual practice, suppress evil and promote good. (*Dongzhen taishang taixiao langshu*, p. 693)

> 凡學上道，大乘之人，修己化世，勿逃山林。山林絕人，中小避欲，合藥試術，研習奇方，是建德之細，非立功之大基。山中立功無所，所以出世市朝，起創治館，繕寫經書，宣行妙法，助國濟時，慈心精勤，抑惡揚善。

Put briefly, what is described in the *Taixiao langshu* reflects the state of the Southern Dynasties Celestial Masters after the internalization of Daoist monastic practice. The old modes of Celestial Master spiritual practice and the new modes of spiritual practice were not mutually exclusive. The premises it describes, built by the Venerable Masters during their practice of Daoism, seem to be a natural transformation of the Celestial Master chambers of quietude and of the halls of parishes. Thus, although it is evident that the *Taixiao langshu* greatly valued celibacy, it did not strictly demand it; it even made a special provision for marital relations between "lay masters" (*zaisu shizi* 在俗師資) and their disciples. Although marriage between a master and an apprentice was forbidden, a master was permitted to marry the daughter of a disciple or recruit a disciple as a son-in-law (*Dongzhen taishang taixiao langshu*, p. 691).

Out of a quest for sanctity, celibacy in Daoist monasteries during the Southern dynasties gradually became a common pursuit. Yet there were no strict limitations in this regard, especially given that there was no rejection of family life in Daoism (Masaharu 1982). Many Daoist monasteries were handed down from master to disciple, while other monasteries, including official monasteries, were passed down to family members. For instance, the Taiping Guan 太平館, built by Emperor Gao of Southern Qi 齊高帝 for Chu Boyu 褚伯玉, was headed by the grandson of Chu's fifth younger brother, Chu Zhongyan 褚仲儼, during the Liang dynasty; a certain Jiang Fuchu 蔣負芻 was in possession of Zhongyang Guan 宗陽館, and later "handed over the day-to-day running of the monastery to his second son, Hongsu 弘素." (*Shangqing Daolei Shixiang* 上清道類事相, p. 877) The family-based inheritance of Daoist monasteries is indicative that Daoist priests were likely to have lived together with their family members, and, moreover, that Daoist monasteries were akin to private family property rather than "the common property" (*changzhu* 常住) of a religious group. In Tao Hongjing's view, almost ten people lived in the very noisy "annexed buldings on the western hill", including Xu Baoguang 徐寶生 and her elder brother Xu Puming 徐普明, her son Zhu Shansheng 朱善生, her nephew Zhou Ziping 周子平, and her servant-girl Lingchun 令春. Most of them belonged to the Xu clan, and the men and women lived together. Although there is no evidence of marital relations between them, it was

very much like a home for them. Michel Strickmann thus argued that: "It would be very wrong to think of Mao Mountain as a truly 'monastic' center. It is clear that the community included both married and unmarried practitioners, as well as large numbers of children." (Strickmann 1978, p. 471) Such circumstances were not at all uncommon during the Southern Dynasties period. Even during the Sui (581–619) and Tang eras, the golden age of Daoist monasticism, the true severing of family ties among Daoist practitioners may not have been strictly enforced. This set the tone for the Daoist monasticism from its earliest days.

## 6. Conclusions

If we agree that the general pattern of the establishment of monasticism originates in radical seclusion and asceticism, that it has led to the development of religious groups with fixed dwellings and the evolution of structured daily life, then this general pattern has parallels with Daoist monasticism (that is, Daoist monastery practices). Yet Daoist monasticism was not a natural outgrowth of the hermetic tradition of withdrawal from secular society. Rather, it underwent a two-stage process of "grafting" in terms of its spiritual beliefs and values. The first stage saw the emergence of the New Daoist scriptures of the Jin and Song periods; in particular, the Lingbao scriptures transformed and distilled the tradition of hermetic practice in the mountains. This stage also borrowed concepts from Buddhism and invested hermitic practice with a more complete and sacrosanct doctrinal foundation. The second stage saw the adoption of and experimentation with the beliefs and values within the Lingbao scriptures by the Southern Dynasties' Celestial Masters order, which introduced the inherent communitarian nature of the Celestial Masters into the development of Daoist monasticism and triggered the large-scale transformation of religious practice among the Celestial Masters of the period. The Celestial Masters' order, which had its roots in popular society, embraced and transformed a mountain-based spiritual practice that had emphasized cutting ties with secular society. This became a key moment in the rise of Daoist monasticism.

To a certain extent, this can explain the regional disparities in the distribution of Daoist monasteries during the Southern and Northern Dynasties: the northern Chinese Celestial Masters order was not initiated into the ways of the "Sandong Jingshu" until almost a century later. The impact of the Lingbao scriptures on the northern Celestial Masters and the existence of the northern Celestial Masters as a group is evident from the large number of Daoist statues that appeared in the late fifth century (Bokenkamp 1997). It is interesting to note that following the introduction of the Lingbao scriptures, the northern Celestial Masters did not widely adopt the model of Daoist monastery practice; the spread of Daoist monasteries in northern China did not occur until after the Sui dynasty. There may have been deeper underlying factors for this related to social structure. In her study of the same geographically distinct phenomenon of Buddist statuary between northern and southern China, Shu-fen Liu 劉淑芬 observes that the Yiyi 義邑 organization, comprised of monks and laymen, was widespread in northern Chinese Buddhism. Southern Dynasties' governments, having implemented a system of guilt by association, exercised a stronger degree of social control, making it harder for grassroots groups to form similar organizations for believers (Liu 2010). This analysis is also applicable to the Daoism of the Southern Dynasties. The organizational structure of the Celestial Masters, based on "mastering the households and ruling the people", was irreconcilably at odds with the state control exercised increasingly by the Southern Dynasties; for southern Daoists, tight grassroots supervision hastened the process of conversion to monastic life. Lu Xiujing's mid fifth-century Daoist reforms grew out of these circumstances, while he himself was the most representative figure in the transformation of the former Celestial Masters into monastery-based practitioners (Wang 2017, pp. 601–706).

In contrast to Christian and Buddhist examples of monasticism, Daoist monasticism has had a unique evolutionary process. Lacking the concept of the other world or original sin, Daoism does not reject secular life, and Daoist monasticism is not so ascetic and extreme,

but rather a communal type of monasticism: dwelling in the mountains without leaving the secular world, living in the monasteries without separating from the family. The Celestial Masters' turn toward monasticism preserved many secular characteristics. Even ascetics who withdrew into the mountains were later disparaged as "people of the Lesser Vehicle" (*xiaocheng zhiren* 小乘之人), and the idea took hold that "superior Daoists practice in the middle of the city, lesser ones do so in distant mountain forests 上士學道在市朝, 下士遠處山林." (*Taishang laojun jiejing* 太上老君戒經, p. 208) In the Southern Dynasties period, many Daoist monasteries were situated in mountain forests, but with the passage of time, more monasteries gradually sprang up in cities (Zhang 2006). Daoist monasteries gradually evolved from being a "home for the Immortals" in the mountain forests into religious service facilities in the metropolises.

**Funding:** This research received no external funding.

**Institutional Review Board Statement:** Not applicable.

**Informed Consent Statement:** Not applicable.

**Data Availability Statement:** No new data were created or analyzed in this study. Data sharing is not applicable to this article.

**Acknowledgments:** This article benefited greatly from the criticisms of four blind reviewers and the translator Damien Kinney.

**Conflicts of Interest:** The author declares no conflict of interest.

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
