# Peer review of "The Celestial Masters and the Origins of Daoist Monasticism"

_religions, doi:10.3390/rel15010083_

Round 1

Reviewer 1 Report

Comments and Suggestions for Authors

This is a very fine article on a historical/religious subject of considerable relevance for China.

The referencing seems excellent to me. The use of primary and secondary sources is very well integrated and you obviously know the material very well indeed. I really like the long quotations from primary sources, with the original given as well.

I also found the level of English-language expression very high indeed. It does not even need very much editing.

I have one suggestion for you to consider. The comparison for true monasticism seems to be St. Anthony. That's fine, but I think it would be useful to give his dates the first time he is mentioned. (He definitely needs to be distinguished from other famous St. Anthonys, such as St. Anthony of Padua.

You say in the conclusion that, by comparison, Daoist monasticism was not "serious" or "thoroughgoing" and you explain why you've reached this verdict, i.e. Daoist monasticism didn't involve total departure from secularism or giving up family.  You also emphasize that its development brought "impurities". I can well see the force of this, but would argue that it might tell us Daoist monasticism was different, rather than not serious. And I'm not sure how to take the term "Chineseize", It seem somewhat pejorative to me. Maybe you intend it as such, but I still think it might be better to frame the whole idea of Daoist monasticism, which is the core of your article, as different from the best-known Christian example of monasticism, rather than not as serious or less thorough-going.

It's just that something rankles by a whole article that is appreciative of the importance of Daoist monasticism, but then ends up implying it isn't serious, not like the Christian equivalent.

This is just a point to consider, not a demand. Your overall attitude is for you to determine, not a reviewer.

Author Response

Thank you very much for taking the time to review this manuscript and for your encouraging and positive comments.

Point 1: I have one suggestion for you to consider. The comparison for true monasticism seems to be St. Anthony. That's fine, but I think it would be useful to give his dates the first time he is mentioned. (He definitely needs to be distinguished from other famous St. Anthonys, such as St. Anthony of Padua.

Response 1: Thank you for pointing this out. I have added his full name and dates of birth and death, as per your comments. (St. Anthony of Egypt, c.251-356)

Point 2: You say in the conclusion that, by comparison, Daoist monasticism was not "serious" or "thoroughgoing" and you explain why you've reached this verdict, i.e. Daoist monasticism didn't involve total departure from secularism or giving up family.  You also emphasize that its development brought "impurities". I can well see the force of this, but would argue that it might tell us Daoist monasticism was different, rather than not serious. And I'm not sure how to take the term "Chineseize", It seem somewhat pejorative to me. Maybe you intend it as such, but I still think it might be better to frame the whole idea of Daoist monasticism, which is the core of your article, as different from the best-known Christian example of monasticism, rather than not as serious or less thorough-going.

Response 2:Your comments on the conclusion part of the article inspired me a lot. I have made rewrites of the key lines of the text as follow: In contrast to Christian and Buddhist examples of monasticism, Daoist monasticism has had a unique evolutionary process. Lacking the concept of the other world or original sin, Daoism does not reject secular life, and Daoist monasticism is not so ascetic and extreme, but rather a communal type of monasticism: dwelling in the mountains without leaving the secular world, living in the monasteries without separating from the family.

Reviewer 2 Report

Comments and Suggestions for Authors

This article explores the rise of monasteries and monastic Daoism during the period of the Southern and Northern Dynasties. Against the background of the Christian and Buddhist tradition, the authors traces the peculiarities of early Daoist monasticism, which did not strictly obligate celibacy or require practitioners to sever ties to secular society. This, the article shows, was to a large part due to a mixing of elements and ideas from both Celestial Master Daoism and what the authors calls Shenxian Daoism (on the latter, see below). The author also provides valuable evidence on the nature, variety and amount of Daoist monasteries by using epigraphic materials. In light of its contribution to the field, I would recommend the article for publication, though I would also suggest a few revisions.

Some general remarks:

I don’t think that the term Shenxian Daoism is used in English scholarship on Daoism and I was not aware of its meaning. When I googled the term, only one result showed up. The designation Shenxian Daojiao in Chinese seems more widespread, but the author should nevertheless clearly explain and delineate at the beginning what is meant by that designation for the general reader. The author may also consider using a different term that is more familiar in English scholarship. The issue of Shenxian Daoism also relates to a more general issue. The authors takes a lot of knowledge on the history of Daoism for granted and specific figures and traditions – including Celestial Master Daoism – may be introduced more thoroughly.

I am not sure that the second section on Patterns of Monasticism adds much value to the article. It is a rather short and generalizing discussion and the points raised therein are only seldomly taken up in the main part of the article, which rather deals with the specific mix of Celestial Master and Shenxian Daoism. The author may want to consider taking out this section or perhaps distilling it into a single paragraph that could be included in the introduction and rather use the additional space to provide more background on the Daoist context of the time as explained above.

Pinyin is not consistent, for example, Zhengao and Zhen’gao, Shenxianzhuan and Shenxian zhuan, Shuijingzhu Jiaozheng, Shuijing zhu Jiaozheng and Shuijing zhu jiaozheng (the last seems correct to me), etc.

Some specific points (and there are more):

Page 2:

“pioneered by Pachomius”; was not yet mentioned and not clear why he is of relevance;

Page 3:
“Ascetic tendencies and behaviors are a permanent fixture of human society.” How so? This should be explained and backed with reference to scholarly literature.

(Thompson 1928): why quote a text that is 100 years old?

Page 5:

“Disaster of Yongjia”: What does this refer to? Should be explained.

Page 7:
It is not clear how “southern Daoism as a whole had already prepared itself for a change of direction toward Daoist monastic practice.”

“psychic medium Yang Xi”: what does this mean, term smacks of esotericism or nineteenth century spiritualism; I think “medium” alone or “medium for the gods” would suffice.

Page 11:
“But very few of them “the Supreme Way” in their practice.” > “But very few of them practice “the Supreme Way.” Same on page 12.

Page 12:

“Thus, it is questionable whether ordinary Daoist monasteries of the Southern Dynasties were able, to a significant extent, to cast aside the former secu-larity of the Celestial Masters and adopt truly monastic practices.” The meaning of this sentence is not clear, please rephrase; also, in how far were Celestial Masters secular?

Page 13:

who study to “Supreme Way,” > “who study the Supreme Way”

Why “Taoism” instead of Daoism here?

„masters in the folklore” (zaisu shizi在俗師資); what is a master in the folkore? zaisu seems to imply a master who has not renounced the ways of the world; the text contains a number of phrases or translations that may not be understandable to the non-specialist

Page 14:

What is an “other house”; features also in other place, I would suggest using a different translation.

Page 15:

Chineseized: sinicized?

Comments on the Quality of English Language

Grammar and language:

The article, though generally well-readable, might profit from another proofreading. I will refrain from mentioning all of the grammatical inaccuracies but just mention a few recurring concerns.

Monastery vs. monastic

Abstract: should be „monastic system“; likewise on the same page “Daoist monastic history”; or: “history of the Daoist monastery”;

Recurring problem: “Daoist monastery phenomenon”; it is not entirely clear what this means, and reads strange from a grammatical point of view; is the author not rather talking about “the emergence/development of the Daoist monastery”?

Reviewer 3 Report

Comments and Suggestions for Authors

Excellent essay!  Well referenced and insightful! It will make a valuable contribution to the field!

Author Response

Thank you very much for taking the time to review this manuscript and for your positive encouragement and comments. 

Reviewer 4 Report

Comments and Suggestions for Authors

This is a decent piece with worthwhile research objectives, and the language is good too. The paper offers an interesting and well-supported conclusion: "Medieval Daoist monasticism was not a serious or thoroughgoing form of monasticism but rather a Chineseized way of monasticism, dwelling in the mountains without leaving the secular world."

I also appreciate that the paper discusses how Buddhism influenced Daoism. Though I recently read another piece discussing the same topic, I am more familiar with people highlighting that Daoism impacted the development of various branches of Chinese Buddhism.

All that said, paper needs some cleaning up.

INTRO 

**The sentence is jumbled, e.g., missing spaces, way too long. Suggest breaking the passage below into at least three sentence:

It has been suggested that virtually all the pre-existing spaces devoted to Daoist practice, such as the temple buildings (jingshe 精舍) of the mountain-dwelling hermits of the Wei (220-266) and Jin (265-420) dynasties, the “chambers of quietude” (jingshi 靖室)of Daoist families, the “halls of parishes” (zhitang 治堂) within the homes of the Celestial Master (tianshi dao 天師道) libationers (jijiu 祭酒), the Mao Mountain 茅山 villa belonging to the Xu family of the Shangqing 上清 school, the temple “abstinence halls” (zhaitang 齋堂) depicted in early Lingbao 靈寶 scriptures, and the guesthouses officially arranged for hermits, are connected with the rise of Daoist monasteries (Kohn 2000; Bumbacher 2000, pp.490-493; Akiko 2009; Wang 2017, pp.3-171; Wei 2017).

***Suggest deleting "which is  incontestable" in the following because it doesn't add anything:

"However, the reasons underlying the rapid spread of Daoist monasteries are perhaps worthy of greater attention than their origins, which are incontestable."

***In the intro, there is an overuse of semicolons, e.g., periods more appropriate in the following: 

"In this paper, I seek to understand the emergence of Daoist monasteries from the perspective of comparative religious history; I also attempt to highlight the role of mountain-based Daoist practice among the Celestial Masters order in the Jin and Song (420-479) periods, which led not only to the establishment of Daoist monasticism, but also to a loss of purity therein."

**Also, it would be good if there's more of a hook.  The content is interesting, but are you, for example, attending to something that a lot of scholars have missed?  If so, it's worth mentioning.

MAIN TEXT

**Your write; "Scholars insist that Daoism is not a monastic religion because it manifests in too many non-monastic ways (Strickmann 1978; Schipper 1984)."  A sentence or two or even a phrase elaborating on what some of these "non-monastic ways" are would be helpful. (I know you go on to explain it, but it would help to have some signposts here

** "Ideally, it would be unproblematic to call Daoist temple practice a form of monasticism. " Strange sentence.  I don't know why "ideally" it is there or really get why it would be either "unproblematic" or problematic.

**A small note, but today Christian orders like the Dominicans are not cloistered if they're men. So you might want to qualify the statement, though your claim about early Christian monasteries seems correct

***suggest replacing "element" with "factor" in the phase "is considered to have been an important element in the rise of monasteries"

**To clarify the direction of the paragraph, I think you should add the bolded text  to the following phrase: "Collective monastic life developed out of hermitic and ascetic practices, characterized by a withdrawal from secular life in organized society"

**Punctuation problems in the sentence that starts with the phrase: "The records in the Shuijing zhu..."

**Ellipses could probably be introduced into some of the longer block quotations in the last third of the paper (there's a lot of pretty long quotes there).

**Throughout, it would be helpful to provide signposts, briefly indicating (e.g., in one or two sentences) the intended purpose of each section at the beginning

Comments on the Quality of English Language

English is pretty good - just some very minor things.
